# Agroforestry as a Biodiversity Conservation Tool in the Atlantic Forest? Motivations and Limitations for Small-Scale Farmers to Implement Agroforestry Systems in North-Eastern Brazil

**Mauricio Sagastuy [1] and Torsten Krause [2],***

[1]   Department of Biological and Environmental Sciences, University of Gothenburg, P.O. Box 461,
40530 Gothenburg, Sweden; 1mauriciosagastuy@gmail.com
[2]   Lund University Centre for Sustainability Studies, Lund University, P.O. Box 170, 221 00 Lund, Sweden
*   Correspondence: Torsten.krause@lucsus.lu.se; Tel.: +46-734239415

**Abstract:** Agroforestry practices support agricultural resilience against climatic variability, increase soil productivity, can diversify and increase farmers' incomes, and support native fauna in agricultural landscapes. However, many farmers are still reluctant to implement agroforestry practices. We distributed questionnaires to 75 agroforestry and 64 "conventional agriculture" small-scale farmers working in the northeastern region of the Atlantic Forest to identify the motivations and limitations to implement agroforestry practices. We reveal the four main reasons why farmers worked with agroforestry: Higher income generation (89%), diversification of the production system (86%), increase in the land's quality and productivity (86%), and increase in self-sufficiency (82%). The three most common mentioned reasons for conventional agriculture farmers to not shift to agroforestry practices were: Uncertainty if the system will work (62%), reduction in yield of the main agricultural crop (43%), and a lack of models and knowledge in the region (41%). Agroforestry in Brazil's Atlantic Forest region can support native fauna, but farmers need to be educated about agroforestry practices and encouraged to switch from conventional agriculture to agroforestry through an increase in available technical assistance and capacitation/training in agroforestry practices.

**Keywords:** cabruca cocoa plantation; crop diversity; small-scale farmers; farmers' perceptions; agroforestry technical assistance

---

## 1. Introduction

### 1.1. Agroforestry for Biodiversity Conservation and Agricultural Production

Agroforestry combines agricultural and silvicultural practices to produce food, wood, and other products [1,2]. Agroforestry systems have been increasingly promoted as land-use systems that can support nature conservation, especially in the tropics [3–5], where it has also been gaining recognition as a tool for reducing poverty, improving food self-sufficiency for farmers, and increasing the productivity and income for small-scale farmers [4,6–8]. Although agroforestry is a potentially more sustainable use of natural resources and land, it is not a "silver-bullet" for reconciling nature conservation and agricultural production [1], because of the irreversible biodiversity value of natural forests.

Two main aspects influence the role of agroforests for biodiversity conservation. First, the species and structural composition of plants in agroforests [9] and second, the degree of the management intensity and human disturbance in these systems [10]. Agroforestry systems that contain a similar

species and structural composition as native forests can be part of a broader regional biodiversity conservation strategy, and potentially serve as buffer zones or ecological corridors [10].

Some of the most important benefits of agroforests for small-scale farmers is the increase in the soil's quality and productivity [6,7], food security [11,12], diversification of the produced goods [7], and more resilience towards pests, market impairs, and climate change [12]. However, small-scale farmers also face challenges. For instance, managing woody perennials and agricultural crops in the same land management unit [13], difficulties in marketing all products [14], and legal limitations such as obtaining licenses to harvest woody perennials [15]. Another limitation for farmers is the delay in return on initial investments, i.e., the time from planting to harvest of produce from trees and woody perennials [9]. In addition, economic, social, and political instability can prevent farmers from investing in longer-term land-use systems (such as agroforestry), because land ownership may not be guaranteed [16].

The benefits and drawbacks related to agroforestry systems mentioned in the last paragraph are commonly known. However, each country and each region of the world face different challenges. To our knowledge, this is the first study that analyzes the social and ecological factors influencing what motivates small-scale farmers to work with agroforestry in the north-eastern region of the Atlantic Forest in Brazil.

The Atlantic Forest of eastern South America is one of the 25 biodiversity hotspots in the world with high levels of biodiversity and endemism [17]. Moreover, it is considered as one of the most threatened forest biomes in the world [18] and one of three most vulnerable biodiversity hotspots to climate change [19]. The Atlantic Forest originally covered around 150 million hectares (ha) along the coast of Brazil and into eastern Paraguay and northeastern Argentina (Figure 1) [20]. Since European colonization and after five centuries of agricultural expansion, industrialization, and urban development, the extent of Atlantic Forest has been reduced to only ~12% of its original extent. Most of the remaining forest cover is distributed in small fragments of 50 ha or less [20]. Only about 9% of the remaining Atlantic Forest is officially protected [20].

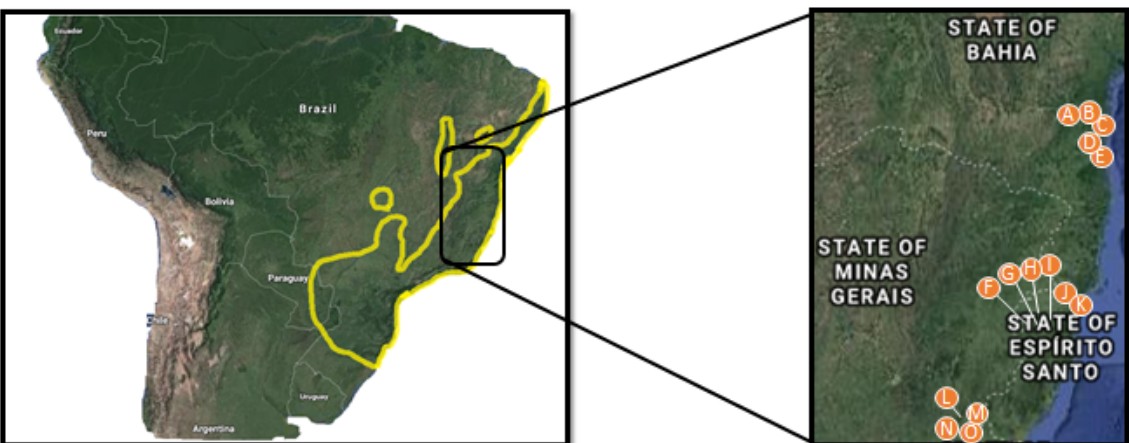

**Figure 1.** Original extent of the Atlantic Forest cover delineated in yellow (left image) and close-up view of the Bahia Sub-Region, which is the study area (right image) (source: Modified from googlemaps.com). The letters in the right map indicate the locations (municipalities) of the farms surveyed in this study. Municipalities where the farms are located: A = Ipiaú, B = Maraú, C = Itacaré, D = Uruçuca, E = Ilhéus, F = Barra de São Francisco, G = São Gabriel da Palha, H = Vila Pavão, I = Nova Venécia, J = Pedro Canário, K = Conceição da Barra, L = Rosário da Limeira, M = São Francisco do Glória, Miradouro, Pedra Dourada, Espera Feliz, N = Muriaé, Miraí, Leopoldina, O = Antônio Prado de Minas, Eugenópolis, Laranjal.

More than 60% of the Brazilian population lives in the Atlantic Forest area (>100 million people) and depend on ecosystem services from the Atlantic Forest biome, for instance water provision [21].

Apart from the motivations of small-scale farmers to work with agroforestry, we inquire about the biodiversity conservation potential of agroforestry in the Bahia Sub-Region (Bahia SR) of north eastern Brazil (Bahia SR according to the biogeographical sub-regions (BSRs) described by Silva and Casteleti [22] based on the main areas of endemism in the Atlantic Forest). The Bahia SR stretches along the coast of Brazil from Sergipe in the north through Bahia and Espírito Santo in the south, extending inland up to 100 miles (a map of the Bahia SR can be found on p.49 in Silvia and Casteleti [22]). In particular, the paper focuses on southern Bahia because of its high endemism of birds, butterflies and primates [22], and its relatively high forest cover characterized by a landscape composed of a mixture of natural forests and shaded cocoa agroforests, locally known as *cabrucas*. Cocoa (*Theobroma cacao*) cultivation began in this region in the eighteenth century and is still an important economic activity in the region [9].

In the late 1980s cocoa production was severely impacted by falling international cocoa prices and the arrival of the fungus *Moniliophthora perniciosa*, causing "witches broom" disease [23]. After this crisis many cocoa farmers sold their shade trees for timber and converted their lands into pastures or planted other crops [24]. Today, the biggest threat to the region's biodiversity and rainforests are the expansion of eucalyptus-monocultures and pastures for livestock [25,26]. Between 2015 and 2016 Bahia was the Brazilian state with the highest rate of deforestation in the Atlantic Forest [27].

Nevertheless, southern Bahia still harbors large forested areas in an agroforestry-native forest mosaic. Over 50% of the land in southern Bahia are forested environments; mainly *cabrucas* (600,000 ha or 26% of the landscape), secondary forests (19%), and primary forests (9%) [28]. These systems are characterized by thinned out native forest, where cocoa is being planted in the understory of the former forest that provides shading for cocoa production and optimal cocoa tree development [9]. In addition, these shade trees are important for protecting the cocoa trees against winds and decrease the risk for biological diseases such as pests, insects, or fungi [29]. Moreover, *cabrucas* play an important role for the conservation of biodiversity in southern Bahia [23].

### 1.2. The Role of Agroforests for the Protection of Emblematic Species in Southern Bahia

Southern Bahia's Atlantic Forest biome is a center of species endemism, containing a great diversity of flora and fauna [9]. A global study carried out by Martini et al., (2007) [30] compared 23 sites known for high arboreal species densities and concluded that southern Bahia had the second highest tree species density in the world. Other studies confirm the high biological diversity found in southern Bahia [9,10,31,32].

Several studies on agroforests in southern Bahia highlight the importance of these systems as habitat for a range of native animal species, particularly given the high degree of forest loss and fragmentation in parts of southern Bahia, where *cabrucas* play an important role in conserving the region's biodiversity [23]. *Cabrucas* provide alternative or additional habitat for several forest species, such as ferns and bromeliads, birds, bats, invertebrates, mammals, leaf-litter herpetofauna, nymphalid butterflies, and other insects; increasing the connectivity between forest fragments, and reducing edge effects to which fragments are exposed [23].

### 1.3. Aims

Since agroforestry systems in the Bahia SR have the potential to support biodiversity conservation and to a certain extent provide habitat for endangered animal species, we analyze the motivations and limitations of small-scale farmers to implement agroforestry systems in their lands. We conducted surveys in the Bahia SR and present insights into the reasons and drawbacks for agroforestry farmers and conventional agriculture farmers to implement agroforestry systems. We inquire to what extent agroforestry can be a viable option for small-scale farmers in the Bahia SR of the Atlantic Forest. Lastly, we ask what barriers exist for small-scale famers and how could these be overcome to increase farmers' willingness to apply agroforestry.

The analysis of agroforestry as a tool for biodiversity conservation focuses in southern Bahia. However, we included a broader range of farmers in the Bahia SR to better understand the motivations and limitations of small-scale farmers to implement agroforestry systems in a wider geographical region.

## 2. Materials and Methods

### 2.1. Farmer Questionnaires

We used structured surveys (in Portuguese) to inquire about farmers' motivations to maintain agroforestry as a predominant land use on their farms, the existing barriers and limitations to continuing agroforestry or implement new agroforestry plots (see Appendix A). The surveys were distributed to small-scale farmers who work with agroforestry and conventional agriculture in southern Bahia, Espírito Santo (which is part of the Bahia SR), and farmers living in the Zona da Mata (Forest Zone), a region in Minas Gerais bordering the Bahia SR. We asked about the different plant and animal species in the lands of the farmers and inquired about available support mechanisms (such as support programs, practical knowledge and examples of the region, monetary incentives, or loans) for promoting agroforestry systems. Lastly, we asked about the motivations, limitations, benefits, risks, and disadvantages associated with the implementation of agroforestry systems.

For the sampling of participating farmers, three local organizations provided us with the logistical support necessary for distributing and carrying out the surveys with farmers in southern Bahia and farmers working in and close to the whole Bahia SR. The three organizations (*Povos da Mata*, *Instituto Capixaba de Pesquisa, Assistência Técnica e Extensão Rural* (INCAPER), and *Iracambi*) who provided logistical support and facilitated the surveys work with agroforestry and conventional agriculture. *Povos da Mata* is a cooperative or association of smallholder farmers working with certification mechanisms of agroforestry systems in the state of Bahia. INCAPER is a state-owned institute working with research, rural extension, and technical assistance related to small-scale agriculture in Espírito Santo. *Iracambi* is a non-profit organization in the Zona da Mata in the state of Minas Gerais and works with biodiversity conservation, research, and the improvement of rural livelihood through forest-based incomes.

The analyzed farmers living in Bahia were members and collaborators of *Povos da Mata*. The farmers from Espírito Santo were farmers that received some kind of technical assistance from INCAPER. The non-profit organization *Iracambi*, contacted a variety of different farmers with the criteria that the farms are located in the Zona da Mata in Minas Gerais. The three organizations who supported this research and carried out the surveys did, however, not specify the response rate for the surveys. Nonetheless, we received responses from 75 agroforestry and 64 conventional agriculture small-scale farmers in the study area. Small-scale farmers are defined according to the Brazilian Atlantic Forest Law [33] as:

Farmers that own a farm not bigger than 50 hectares, and work their farms with their own personal work and with the work of their family, with the eventual help of third parties, (also collectives where the area of the land per person is not bigger than 50 hectares), and at least 80% of the gross income of the farm has to come from activities related to agriculture, livestock or silviculture or from rural extractivism.

The categorization of farmers working with agroforestry or conventional agriculture is based on whether or not the farmers have woody perennials and agricultural crops and/or animals on the same land unit. The farmers that have woody perennials on the same land unit as agricultural crops and/or animals where classified as "agroforestry" respondents and the farmers that did not were classified as "conventional agriculture" respondents. The survey is provided in Appendix A.

### 2.2. Description of Form and Development of the Surveys

We used a survey with closed-end responses (the respondents were also given a free line to express extra answers, thoughts or factors related to the questions. See Appendix A), based on a methodology

and set of questions developed by the World Agroforestry Centre [15]. Moreover, we consulted the three mentioned organizations and piloted the survey to ensure the questions were adequate and logical, allowing us to collect data on the motivations and limitations for farmers to create agroforestry systems on their lands and to inquire how to support and enhance farmers' willingness to convert to agroforestry.

### 2.3. Analysis of the Questionnaires

We provided a definition of agroforestry and a conventional agriculture farmer to the three organizations that supported us with the distribution of the questionnaires. However, the final classification and selection of farms was done by the three organizations we worked with (*Povos da Mata*, INCAPER, and *Iracambi*). We validated the classification carried out by these organizations through an analysis of the data on the average number of plant species on the individual farm's land unit, confirming that agroforestry farmers had substantially more plant species (around six different plant species) in their farms compared to conventional agriculture farmers (around two different plant species per farm) (Figure 2).

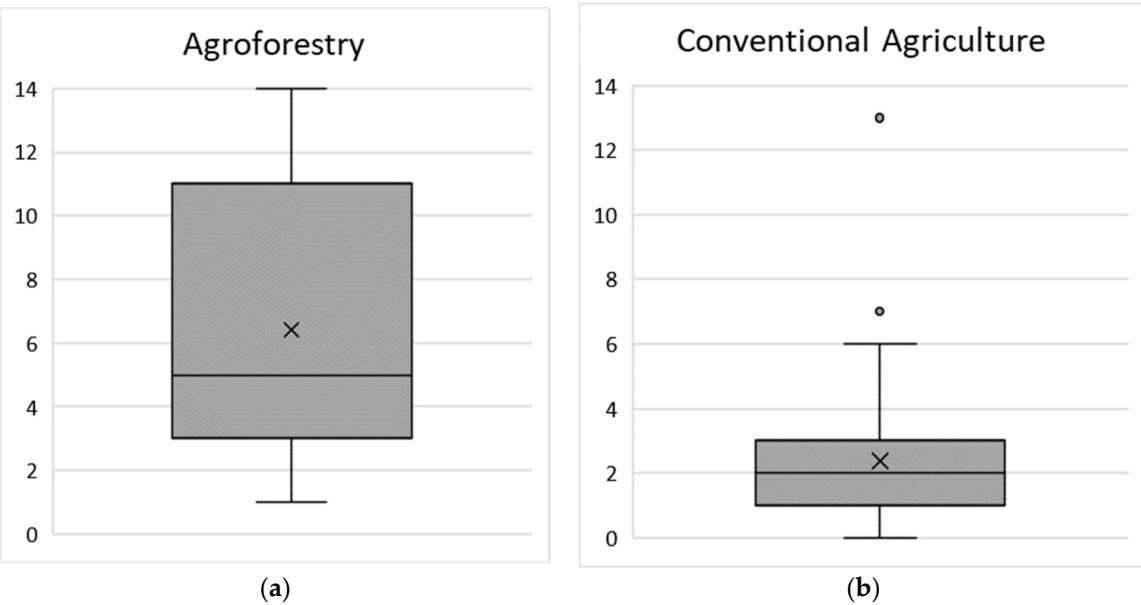

(**a**)                               (**b**)

**Figure 2.** Boxplot showing the number of plant species per farming system. The × represents the average number of plant species, the line inside the box represents the median, and the dots outside the box represent the outliers. Agroforesty (**a**) n = 75, conventional agriculture (**b**) n = 64.

The nature of the three different organizations that contacted the farmers potentially influenced the type of farmers that took part during this study. For example, *Povos da Mata* is an organic certification group and most of the agroforestry farmers were part of this organization, representing a selection bias as it is likely that these farmers hold more environmentally friendly views than average farmers. The organization that contacted farmers living in Espírito Santo (INCAPER) is a state-owned institute working with research, rural extension, and technical assistance related to a variety of agricultural practices. The farmers contacted through this organization were usually farmers that received some kind of technical assistance from INCAPER. The non-profit organization *Iracambi* that is located in the Zona da Mata in Minas Gerais, contacted a variety of different farmers with the criteria that the farms are located in the Zona da Mata in Minas Gerais. The three organizations provided the questionnaires face-to-face and ensured that the farmers that were not literate were able to participate in the study.

From the 75 agroforestry farmers that took part in this study the majority are located in southern Bahia (57%), followed by agroforestry farmers living in the Zona da Mata in south-eastern Minas Gerais (30%) and Espírito Santo (12%). From the 64 interviewed conventional agriculture farmers

67% were located in the Zona da Mata in south-eastern Minas Gerais, followed by Espírito Santo (28%), and southern Bahia (5%). The different states are shown in Figure 1. Moreover, Appendix B contains two different tables with the locations of each farm and the farms' sizes (for agroforestry and for conventional agriculture respectively).

## 3. Results

### 3.1. Characteristics of the Two Groups of Farmers

Only small-scale farmers with land ranging from 1 to 50 hectares were selected and differentiated into agroforestry farms and conventional farms. The average number of plant species per land unit (43 agroforestry and seven conventional agriculture farmers mentioned "hortaliças", which translates to "vegetable gardens" as a category of plants on their farms. We counted hortaliças as one plant species) in agroforestry systems was around 6.4 species per farm, and there was a range between 1 to 14 different plant species per farm (Figure 2a). The two most commonly used livestock on agroforestry farms were chickens (19% of farms) and cattle (16%), followed by beekeeping (4%) and aquaculture (4%).

In conventional agriculture the average number of plant species per land unit was approximately 2.4 species per farm, and there was a range between 0 to 13 different plant species per farm (Figure 2b). The three most common livestock on conventional agriculture farms were cattle (present in 44% of the farms), pigs (6%), and chickens (5%).

### 3.2. Farmers Surveys: Animal Species Presence on Agroforestry Farms

We compared the presence of different animal groups on agroforestry farms (75 responses) and conventional agriculture farms (64 responses) (Figure 3). According to the responses from farmers, all of the animal groups we selected were more commonly found in agroforestry systems than in farms working with conventional agriculture. The largest difference between both systems was found in the category of "other large mammals" (e.g., peccary, deer). While 52% of the farmers working with agroforestry systems confirmed the presence of other large mammals in their lands, only 8% of conventional farms did. In addition, most farmers (93% agroforestry (n = 74) and 87% conventional agriculture (n = 63)) stated that they like to have wild animals on their lands.

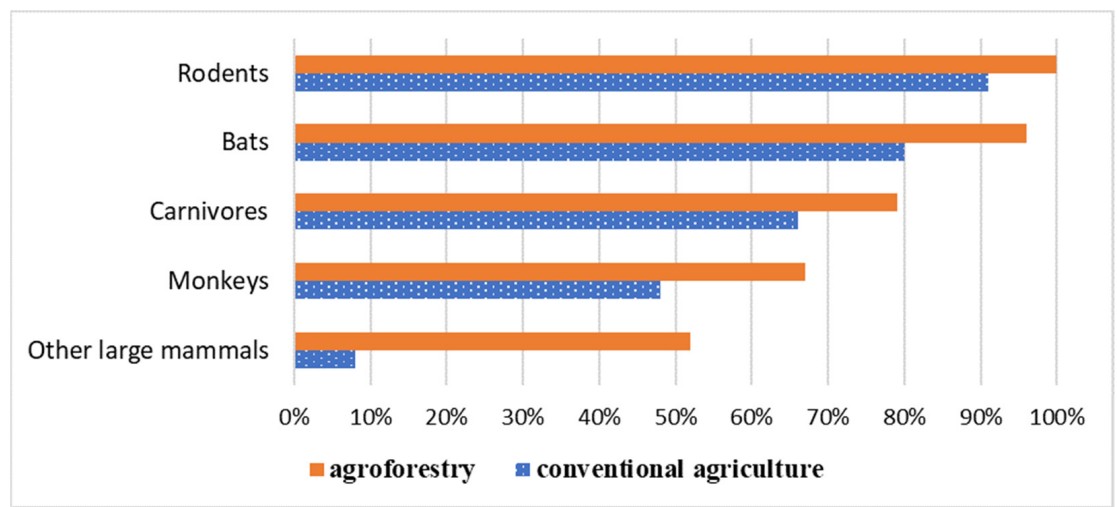

**Figure 3.** Percentage of farms with animal species present in agroforestry systems (n = 75) and conventional agriculture (n = 64), based on farmers' answers.

When classifying the presence of different animal groups according to the farming practice and the state where the farm is located, one can see that the agroforests in southern Bahia are the systems

most frequently visited by the different animal groups (Figure 4). Moreover, even when analyzing the different farming systems according to each state, agroforests are usually more often visited by the different animal groups compared to conventional farms in each state. In Figure 4 only three conventional agriculture farmers from southern Bahia are included, which is not representative for conventional farmers in southern Bahia.

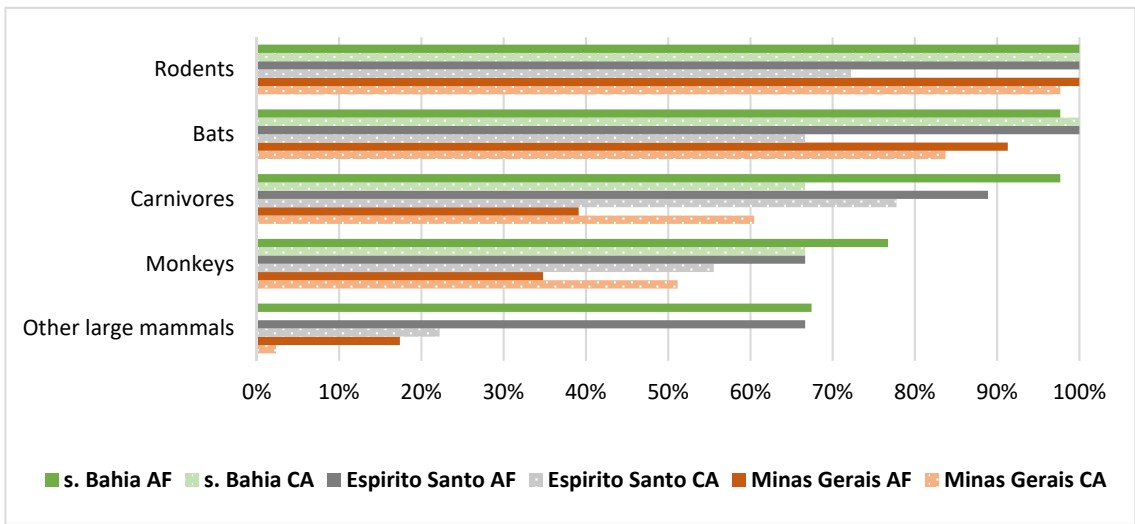

**Figure 4.** Percentage of farms with animal species present in agroforestry systems from southern Bahia (n = 43), Espírito Santo (n = 9), and Minas Gerais (n = 23) and conventional agriculture farms from southern Bahia (n = 3), Espírito Santo (n = 18), and Minas Gerais (n = 43). The results of this Figure are based on farmers' answers. AF means agroforestry farmers, and CA means conventional agriculture farmers.

*3.3. Agroforestry Versus Conventional Agriculture—Motivations and Limitations for Small-Scale Farmers*

We inquired whether farmers are aware of and participate in the most commonly used agricultural and agroecological governmental support programs for the Atlantic Forest region [15] (Figure 5). PRONAF (Programa de Fortalecimento da Agricultura Familiar) is one of Brazil's most well-known agricultural support programs for small-scale farmers and provides low interest loans so that farmers can invest in agricultural or agroecological practices [15]. The PAA (Programa de Aquisição de Alimentos) is a governmental support program, where small-scale farmers sell their food products directly to governmental institutions that work with food security in Brazil (such as food banks, community kitchens, or institutions that work with social aid) [15]. The current PNAE (Programa Nacional de Alimentação Escolar) is a governmental support program (created in 2009) that demands public schools in Brazil to buy at least 30% of the school's lunch directly from regional small scale farmers [15]. Figure 5 shows that agroforestry farmers are more aware of and participate more in the PAA and PNAE support programs compared to conventional agriculture farmers. On the other side, conventional agriculture farmers are more aware of and participate more in the PRONAF support program.

Conventional agriculture farmers were asked whether they would like to intercrop with woody perennials on their fields (in other words implement agroforestry in their land). Out of the 64 conventional farmers surveyed, a slight majority of 55% was positive to agroforestry farming practices on their lands. The main benefits conventional farmers perceived with agroforestry were an increased quality and productivity of the land (77%), conservation of regional biodiversity (69%), and the ability to produce products at different times of the year (51%) (Figure 6).

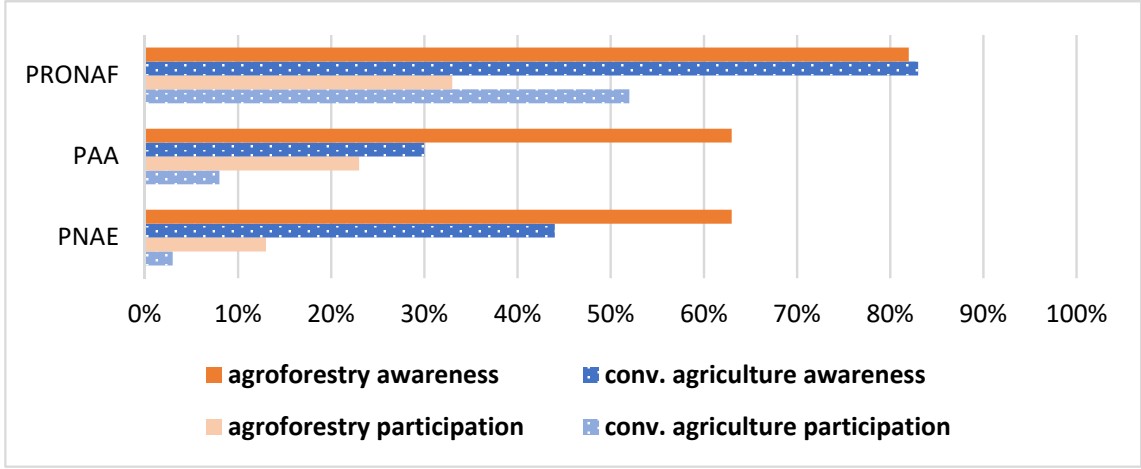

**Figure 5.** Awareness of the existence of and participation in governmental agricultural/agroecological support programs (multiple responses possible). In dark and light orange are the responses of the agroforestry farmers (n = 75) and in light and dark blue are the answers of the conventional agriculture farmers (n = 64). PRONAF, Programa de Fortalecimento da Agricultura Familiar; PAA, Programa de Aquisição de Alimentos; PNAE, Programa Nacional de Alimentação Escolar.

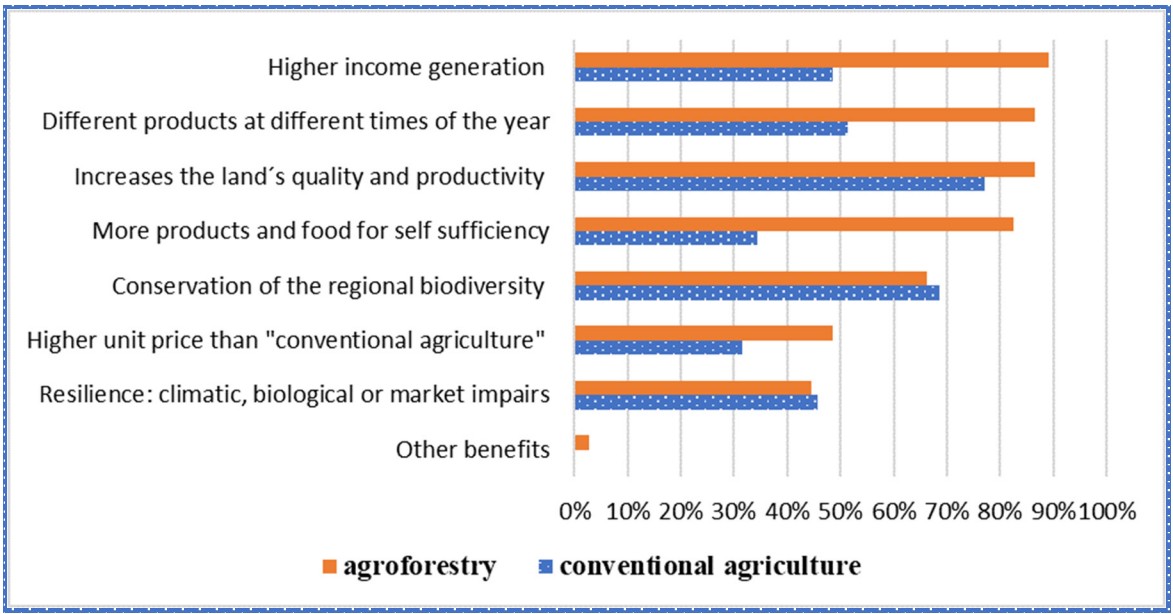

**Figure 6.** Main reasons and benefits of working with agroforestry compared to conventional agriculture by agroforestry farmers (n = 74). The expected benefits of switching to agroforestry practices, mentioned by the conventional agriculture farmers (n = 35) that would be willing to switch to agroforestry practices in their lands.

The perceived benefits for agroforestry farmers to work with agroforestry compared to working with conventional agriculture were higher income generation (89%), the ability to produce different products at different times of the year (87%), and an increase in the land's quality and productivity (87%). Overall, agroforestry farmers are more aware of the benefits of working with agroforestry compared to conventional farmers (Figure 6). In Appendix C we show the results of Figure 6 classified according to where the farms are located: Southern Bahia, Espírito Santo, or Minas Gerais.

Agroforestry and conventional agriculture farmers recognize similar limitations of working with agroforestry (Figure 7). The three most important limitations for conventional agriculture farmers to switch to agroforestry were: Uncertainty if the system will work (62%), reduction in yield of the main agricultural crop (43%), and a lack of models and knowledge in their region (41%). Limitations most

frequently mentioned by agroforestry farmers were the long production time for woody perennials (39%), but also the reduction in the yield of the main agricultural crop (34%). In the category "other limitations", agroforestry farmers mentioned a lack of extra free space for planting (16%) and a difficult access to get seedlings in their region (7%) as important disadvantages. In the same category, conventional agriculture farmers mentioned the following aspects as the most important in the category: A lack of water for planting woody perennials (5%) and the lack of extra free space for planting (3%).

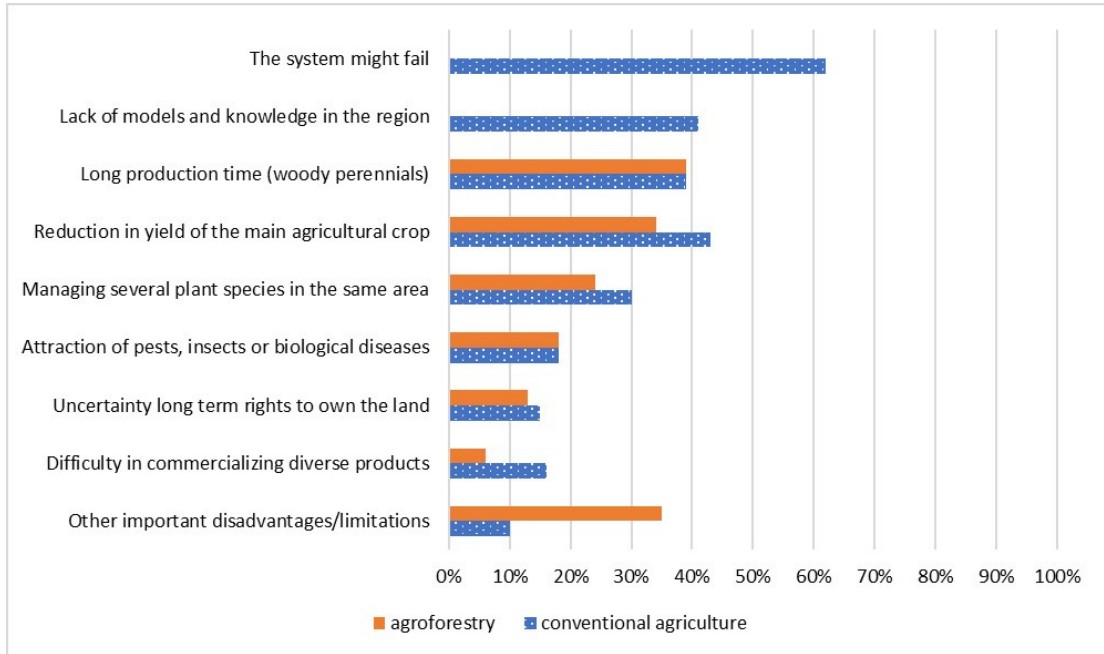

**Figure 7.** Perceived limitations and drawbacks of working with agroforestry (agroforestry farmers n = 62). Expected limitations and drawbacks related to the implementation of agroforest systems in the lands of conventional agriculture farmers (conventional farmers n = 60). The first two categories in the results ("the system might fail" and "lack of models and knowledge in the region") were only included in the survey to conventional agriculture farmers.

Figure 8 shows the limitations and disadvantages mentioned by the farmers in Figure 7, but in Figure 8 the answers were classified according to the three analyzed regions. In Figure 8 there were only three conventional agriculture farmers from southern Bahia and only three agroforestry farmers from Espírito Santo. The disadvantages of "uncertainty long term rights to own the land" was only mentioned by farmers living in Minas Gerais (40% agroforestry and 22% conventional agriculture farmers).

In Figure 9 we classify "other important disadvantages/limitations" of working with agroforestry, mentioned in Figures 6 and 7, according to the three different states. A lack of extra free space is the problem most often mentioned by all farmers, especially for farmers working in southern Bahia (24%). Moreover, a lack of water is a limitation only mentioned by farmers living in Espírito Santo (16%).

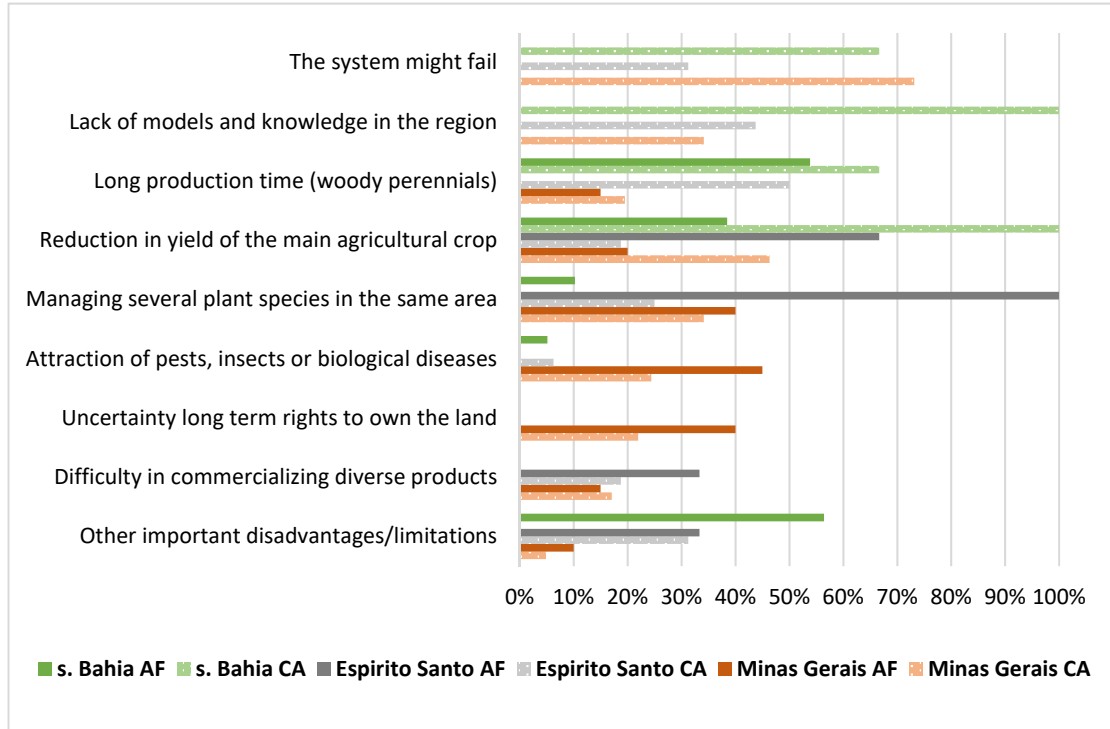

**Figure 8.** Perceived limitations and drawbacks of working with agroforestry, answered by agroforestry farmers from southern Bahia (n = 39), Espírito Santo (n = 3), and Minas Gerais (n = 20). Expected limitations and drawbacks related to the implementation of agroforest systems in the lands of conventional agriculture farmers from southern Bahia (n = 3), Espírito Santo (n = 16), and Minas Gerais (n = 41). The first two categories in the results ("the system might fail" and "lack of models and knowledge in the region") were only included in the survey administered to conventional agriculture farmers. AF means agroforestry farmers, and CA means conventional agriculture farmers.

| Other important disadvantages/limitations | s. Bahia | Espírito Santo | Minas Gerais |
|---|---|---|---|
| Lack of extra free space | 24% | 5% | 3% |
| Difficult access to get seedlings | 7% | 0% | 5% |
| Lack of water | 0% | 16% | 0% |

**Figure 9.** Other important limitations and disadvantages of working with agroforestry according to both agroforestry and conventional agriculture farmers from southern Bahia (n = 42), Espírito Santo (n = 19), and Minas Gerais (n = 61). Other constraints with a low number of respondents mentioning them were: Lack of "additional" family labor; inability to plant trees in "poor" soils; trees casting too much shadow; pruning, planting, and managing trees requires more work; lack of time; more focus on the vegetables/crops due to a faster investment return; lack of money; difficulty in commercializing agroforestry products.

## 4. Discussion

### 4.1. Influence of the Regional Distribution of the Farmers in the Results

The geographical distribution of our respondents with agroforestry and conventional agriculture farms across the three states (see methods section) explains some of the variations in some of our results.

For example, the stated visits of each animal group in agroforestry systems compared to conventional agriculture is shown in Figure 3. Several studies reveal a higher wild animal species diversity in agroforestry systems [23,32,34]. Our results support this and show that the plant diversity and structural composition of agroforestry systems makes these systems more suitable as habitat or corridor by a more diverse group of animal species as compared to conventional agriculture. However,

southern Bahia has more forest cover compared to Espírito Santo or south-western Minas Gerais, and since most agroforestry farmers we surveyed are located in southern Bahia, more wild animals would be expected there, independent of the type of farm. Nonetheless, Figure 4 shows that the different animal groups usually visited more agroforest systems than conventional farms even when differentiating the farming system according to the state where the farms are located. The differences presented in Figure 4 are not pronounced and other factors are likely to confound and influence these results (such as the undisturbed forests surrounding the farms).

*4.2. Agroforestry as a Tool for Biodiversity Conservation*

4.2.1. Species Presence and Conservation in Agroforests

Since southern Bahia still has a relatively high forest cover, *cabrucas* and other agroforestry systems have the potential to serve as part of a regional biodiversity conservation strategy [28]. However, *cabruca*-dominated landscapes contain fewer biological communities compared to forest-dominated areas [23]. Large forest fragments are needed for biodiversity conservation and *cabrucas* can, at best, support landscape conservation efforts. Yet, a landscape dominated by agroforests and with few undisturbed habitats is unlikely to retain the original species assemblages over a long term. Therefore, although agroforestry systems in southern Bahia were more often visited by different animal groups (Figure 4), they alone cannot provide a good enough habitat for these species and other local biodiversity [23]. Protecting the remaining areas of Atlantic Forest fragments is crucial as the presence of undisturbed natural forests is the most important factor for conserving the region's remaining biodiversity [23].

As mentioned earlier in the discussion, agroforestry systems in southern Bahia were the system that was most often visited by a range of animal groups (Figure 4). The agroforestry systems in Espírito Santo where almost as often visited by the different animal groups as the agroforests in southern Bahia (Figure 4). The least often visited agroforestry system where the ones located in Minas Gerais (Figure 4). This raises the question whether the agroforests in southern Bahia where so often visited by wild animals, because they are so highly diverse and structurally complex or is it because southern Bahia is a region that is more forested than Espírito Santo and Minas Gerais? Likewise, are the agroforest systems in Minas Gerais simply less biologically rich? Or are they less frequently visited by wild animals due to the lower forest cover in Minas Gerais? According to Cassano et al. [10] the proximity to undisturbed forests and the management intensity in *cabrucas* affect the conservation value and the presence of different mammals in these agroforestry systems. They characterized management intensification in *cabrucas* by the direct effect of reduced canopy cover and the indirect effect of higher frequency of dogs. Cassano et al. [10] concluded that even if both factors (proximity to forests and management intensification) affected the distribution of mammal species, management intensification influenced the presence of mammals the most, negatively affecting a larger number of species. However, the importance of local forest cover and local management intensification are likely to be context dependent. For example, Pardini et al. [35] showed that in the region of the Serra do Mar in the southern part of the Atlantic Forest, the abundance and richness of specialized mammals increased with forest patch size in a landscape containing an intermediate proportion of remaining forest (30%), but not in more forested (50%) or deforested (10%) landscapes.

4.2.2. Farmers' Willingness to Support Biodiversity Conservation through Agroforestry

Agroforestry can be part of a conservation strategy for a range of animal species [23,34]. However, the attitudes of farmers towards biodiversity and their conservation are crucial and it is important to understand if farmers are willing to support local biodiversity and adapt their farming practices accordingly. Our findings suggest that the vast majority of both groups of farmers have a positive attitude towards wildlife on their lands, which is important for using agroforestry systems as

part of a wider biodiversity conservation strategy serving as buffer zones or biological corridors in the area.

Most farmers we surveyed like to have wild animals in their lands. Yet, it is important to understand the possible management practices farmers can implement in order to promote biodiversity conservation without affecting the production in their lands. In the context of our study, controlling domestic dog populations and discontinuing hunting are two actions that directly support the abundance of wildlife and that farmers could implement without actually changing their land use [10,32]. Although hunting is illegal throughout the Atlantic Forest it is not controlled by the public authorities [23]. Moreover, hunting is increasingly becoming a leisure activity replacing hunting for subsistence [31]. Canopy connectivity also plays an important role for promoting the occurrence of arboreal animals in agroforestry systems, but increasing shade cover could affect the productivity of cocoa and other crops [10]. To reconcile productivity and yields with canopy connectivity, it is suggested to plant canopy corridors and maintain specific tree species instead of increasing overall tree densities throughout the agroforestry system [10]. This allows arboreal species to move easily between different forest patches, boosting local populations and avoiding further fragmentation of habitats.

*4.3. Agroforestry Versus Conventional Agriculture—Motivations and Limitations for Small-Scale Farmers*

4.3.1. Comparing the Economics of Agroforestry Systems and Conventional Agriculture

The main reason why farmers work with agroforestry was higher income, mentioned by 90% of the farmers working with agroforests as one of the main benefits (Figure 6). A number of studies in Brazil also state that agroforestry systems can generate higher revenues than conventional agriculture in the medium to long term [36,37]. However, the profitability of an agroforestry system or a conventional agriculture farm is context dependent and factors such as crop species used, maturity of the system, labor and material inputs, management practices, market prices for produce, certification mechanisms, and climatic or biological stress and/or disturbance affect the farm incomes [38,39]. Performing a more comprehensive economic analysis with the inclusion of other parameters such as labor and material inputs, plant species and plant density, maturity of the system, management practices (such as the use of machinery, fertilizers, pesticides), purpose of the farm (such as erosion control, supply of products for self-sufficiency, provision of ecosystem services), and soil characteristics is necessary to generate more reliable data.

4.3.2. Farmers' Perceptions of Agroforestry and How to Increase Conventional Farmers' Willingness for Agroforestry

According to the agroforestry farmers who responded to the survey, the economic benefits of agroforestry is the main reason why they chose to work with this type of system (Figure 6). However, there are three other important reasons that determine why farmers chose to work with agroforestry. Agroforestry is seen as a more holistic production system by agroforestry farmers compared to conventional agriculture farmers (Figure 6). Agroforestry farmers perceive agroforests as a farming system that provides several important benefits besides higher income generation. These benefits are higher product diversity, higher land productivity, and increased self-sufficiency (Figure 6). On the other hand, conventional agriculture farmers do not seem to value product diversity and an increase in self-sufficiency as important benefits derived by agroforestry systems and they also perceive less economic advantages in working with agroforestry compared to agroforestry farmers (Figure 6).

To increase farmers' willingness to switch to agroforestry it is crucial to provide access to technical assistance, rural extension, and capacitation/training in agroforestry practices. Two out of the three most important limitations (the system might fail, and lack of successful models and knowledge in the region) (Figure 7) can be addressed by showcasing successful examples of agroforestry systems in the region and by providing capacity training on agroforestry systems. The conventional agriculture

farmers who participated in the study associate a range of benefits to agroforestry systems and about 50% of respondents associate it with higher income generation. Moreover, more than half of the conventional agriculture farmers responded that they would like to work with agroforestry in their lands, or at least consider it. Showcasing successful agroforestry models and supporting research into its application and practical knowledge into how to create these systems could help to increase the number of conventional agriculture farmers willing to shift to agroforestry systems.

Moreover, agroforestry systems usually have higher establishment costs and the investment returns often come in the medium to long term [36]. Figure 7 shows that agroforestry and conventional agriculture farmers perceive the long production time of woody perennials and the reduction in yield of the main agricultural crop as important disadvantages of working with agroforestry. Additional subsidies for the initial switch from conventional agriculture to agroforestry could further increase farmers' willingness for agroforestry. These subsidies could come from governmental support programs (such as the ones listed in Figure 5) or by payment for ecosystem services programs. The payment for ecosystem services is a tool often mentioned in the scientific literature that can help farmers in overcoming the initial higher establishment costs associated with agroforestry systems [39,40].

Moreover, it is evident that each region has different challenges related to the adoption of agroforest systems (Figures 8 and 9). For example, Figure 8 shows that the limitation of "uncertainty long term rights to own the land" is a problem only mentioned by farmers working in Minas Gerais (40% agroforestry and 22% conventional agriculture farmers). Thus, more secure land rights can support small scale farmers in the creation of agroforestry systems in Minas Gerais. Moreover, a "lack of extra free space" is a limitation that was often pointed out by farmers living in southern Bahia (24%) and a "lack of water" is a disadvantage mentioned only by farmers living in Espírito Santo (16%) (Figure 9). Understanding how these and other factors affect the adoption of agroforestry systems by small-scale farmers in each region is crucial in order to provide accurate information for policy makers and private actors to support small-scale farmers in adopting or maintaining agroforestry practices on their lands.

## 5. Conclusions

Our research demonstrates the importance of analyzing farmers' motivations and limitations from different perspectives (farming practices, location of the farm). We share important insights of the complexity of how small-scale farmers make decisions related to the adoption of agroforestry practices. We presented local knowledge and perceptions about agroforestry provided by the two sets of farmer groups, those who practice agroforestry and those who use more conventional agricultural practices. The selected agroforestry farms contain a higher number of plant species than conventional agriculture farms and according to farmers were frequented by a larger variety of wildlife. In line with other studies, our findings reveal that agroforestry in the Atlantic Forest has the potential to serve as buffer zones around forest remnants and protected areas, and as ecological corridors between forest remnants. Nevertheless, although agroforests can under certain circumstances serve as part of a wider biodiversity conservation plan, they do not replace the conservation value of natural forests.

Agroforestry farmers perceived more benefits of working with agroforestry than conventional agriculture farmers. Agroforestry farmers state that agroforestry is a more holistic production system than conventional agriculture and according to them the main benefit is higher income generation followed by an increase in product diversity, productivity, and self-sufficiency.

More than half of the conventional agriculture farmers who responded to our survey expressed that they would consider working with agroforestry on their lands. Improving technical assistance, rural extension, and capacitation/training in agroforestry practices is necessary in order to address the main limitations for conventional agriculture farmers to start implementing these systems in their lands and increase farmers' willingness for agroforestry. In addition, a growing number of successful agroforestry examples would further facilitate the opportunities for conventional farmers to switch. Expanding and including more farmers into programs that pay for the ecosystem services provided by their lands may further promote the use of sustainable farming practices and help in overcoming

the initial economic obstacles in switching to agroforestry [39,40]. Nonetheless, in that respect it is important to understand why farmers, who are generally well aware of the different governmental support mechanisms, are not already participating in these (Figure 5).

Furthermore, we also suggest that future studies along geographic, biodiversity, and crop/ produce boundaries would be an important contribution to disentangle the complexity surrounding different types of farms and farmers' willingness to continue or switch to agroforestry and thereby support regional biodiversity.

**Author Contributions:** Conceptualization, M.S. and T.K.; data curation, M.S.; formal analysis, M.S.; investigation, M.S.; methodology, M.S.; project administration, M.S. and T.K.; resources, T.K.; supervision, M.S. and T.K.; validation, T.K.; visualization, M.S. and T.K.; writing—original draft, M.S.; writing—review & editing, M.S. and T.K.

**Funding:** Torsten Krause received funding from the Swedish Research Council for his involvement during the research project (Grant 2016-00583, Torsten Krause). The Gothenburg University funded the distribution of the questionnaires to the farmers.

**Acknowledgments:** We would like to acknowledge Johan Uddling (senior lecturer from the University of Gothenburg) for his support during the research and his comments on the manuscript. It is also important to point out that this research would not have been possible to execute without the kind participation of the three organizations working in the north-eastern region of the Atlantic Forest: *Povos da Mata*, INCAPER and *Iracambi*. We appreciate the generous and informed involvement from the contacts of each organization: Nicole Lelllys from *Povos da Mata*, Maria da Penha Padovan from INCAPER, and Jose Luiz Barbalho de Mendoça from the *Iracambi Research Center*. Moreover, we would like to thank the kind participation and valuable information given by every farmer that took part in this investigation.

**Conflicts of Interest:** Authors declare no conflict of interest. The funders had no role in the design of the study; in the collection, analyses, or interpretation of data; in the writing of the manuscript, or in the decision to publish the results.

## Appendix A. Questionnaire for Farmers Working with Agroforestry and Conventional Agriculture

### Questionnaire

Dear Sir or Madam, thank you for agreeing in taking part on this research questionnaire. My name is Mauricio Sagastuy. I am a student at the Gothenburg University in Sweden. 3 years ago, I had the opportunity to work in a rainforest reserve in southern Bahia. This experience triggered my interest and devotion to sustainable land use practices in the Atlantic Forest of Brazil. This questionnaire is the essence of my Master thesis. In my research project I aim to answer two questions for the north-eastern Atlantic Forest biome: first, to what extent can agroforest systems provide habitat for key animal species? And second, what are the motivations and limiting factors for family farmers to create agroforest systems? With this questionnaire I want to hear directly from the farmers their experiences and thoughts related to subjects about conventional agriculture and agroforestry. Therefore, with the data given to me in the questionnaire, I will be able to answer the second question of my research project.

The participation in this questionnaire is entirely voluntary and you are not obliged to answer these or any question if you don't want to or feel uncomfortable with it. All the information will be used for non-commercial purposes. Also, be assured that the information you provide will be kept in the strictest confidentiality and will be kept anonymous. This questionnaire should only take 15 min to complete.

*INFORMATION ABOUT THE FARM AND THE CURRENT PRACTICES (FOR BOTH KIND OF FARMERS)*

*Please fill in the information below and check with an X the boxes that are correct/fit to you and your farm. You can check more than one box per question*

Which of these organizations provided you the questionnaire?

&#9633;    Povos da Mata

&#9633;    INCAPER

&#9633;    Iracambi

A)    <u>General information about the farm:</u>

- Name of the owner: _______________________________________________
- Name of the farm: _______________________________________________
- Location (state and municipality): _______________________________________________
- Size of the farm (in ha): _____________________
- Area used for agricultural/agroforestry purposes in your farm (in ha): _______________
- How far is your farm to the next road (in meters or kms): _______________
- How far is your farm to the closest "primary or secondary forest" (in meters or kms) _______________________

- Do you have an area where you plant products for your own/family consumption?

  &#9633;    Yes

  &#9633;    No

- Mention the plant and animal species on your agricultural land (excluding the land you use for own consumption). In other words, mention the species/products you sell/use for commerce.

  (*It can be that you work with just one or that you work with more species*):

  1.
  2.
  3.
  4.
  5.
  6.
  7.
  8.
  9.
  10.

B)    <u>Socio-environmental data:</u>

- Do you consider your soil fertile?

  &#9633;    yes

  &#9633;    no

- Do you irrigate your land?

  &#9633;    Yes

  &#9633;    No

- Do you take or took part in any of the following support programs?

  &#9633;    Technical assistance and rural extension

  &#9633;    Capacitation/training for your current agricultural practice

  &#9633;    I am part of an agricultural/agroecological organization, institution or support group

C)　　Income per hectare in the last 3 years:

- What was approximately your monthly income per hectare in the last 3 years (in Brazilian reals) If you know the information just for some of the years you can write that down too:

| Year | Monthly Income Per Hectare |
|------|----------------------------|
| 2017 | |
| 2016 | |
| 2015 | |

*QUESTIONS FOR BOTH KIND OF FARMERS*

**Questions about governmental programs that can support the creation of agroforest or agricultural systems**

1. **Do you know of the existence of any of the following governmental programs that promote or can support the creation of agricultural or agroecological systems?**

   *If yes, please mark the programs that you are aware of.*

   Programa Nacional de Alimentação Escolar—PNAE

   ☐　　Programa de Aquisição de Alimentos—PAA
   ☐　　PRONAF - Programa Nacional da Agricultura Familiar
   ☐　　Payment for ecosystem services—Atlantic Forest Law
   ☐　　Other governmental programs: _______________________________

2. **Do you take or took part in any of the following governmental programs?**

   *Please mark the programs you take or took part of.*

   ☐　　Programa Nacional de Alimentação Escolar - PNAE
   ☐　　Programa de Aquisição de Alimentos - PAA
   ☐　　PRONAF - Programa Nacional da Agricultura Familiar
   ☐　　Payment for ecosystem services -Atlantic Forest Law
   ☐　　Other governmental programs: _______________________________

**Questions about the acceptance of wild animals**

3. **Do some of the wild animals listed below come through or live on your land?**

   *If yes, please mark the animals that come through or live in your land.*

   ☐　　Maned sloth (just for the farmers living in southern Bahia)
   ☐　　Golden-headed-lion tamarin (just for the farmers living in southern Bahia)
   ☐　　Golden-bellied capuchin (just for the farmers living in southern Bahia)
   ☐　　Other monkeys
   ☐　　Carnivores (such as oncilla, ocelot, crab-eating fox, and other carnivores)
   ☐　　Other large mammals (such as peccary, deer)
   ☐　　Rodents (such as mice, squirrel, bristle-spined rat and other rodents)
   ☐　　 Bats

4. **Do you accept/like having wild animals on your land (such as monkeys, small or large rodents, foxes, armadillos, and other small or large animals)?**

   ☐ Yes
   ☐ No

   4.1 **If your answer was yes, what are the reasons for you to accept/like having these wild animals in your land??**

   ☐ They can act as "regulators" on my land and thus avoid biological diseases/ making my system more resilient
   ☐ Animals have a right to live here too
   ☐ I can hunt them and use as food source
   ☐ Other: _______________________________________________

   4.2 **If your answer was no, what are the reasons for you not to accept/like having these wild animals on your fields?**

   ☐ They can damage my crops/domesticated animals
   ☐ They can damage my household and the items in it (such as food, pets, etc.)
   ☐ They can bring pests and diseases to the land
   ☐ Other: _______________________________________________

**Questions about intercropping and agroforestry**

5. **What do you think/is your perception about intercropping woody perennials with your main agricultural crop?**

   *Please mark with an X, the answers that you agree with:*

   ☐ It can decrease the income and the production of the land in the short term
   ☐ It can decrease the income and the production of the land in the long term
   ☐ It can increase the income and the production of the land in the short term
   ☐ It can increase the income and the production of the land in the long term
   ☐ It can bring more pests and diseases to the system
   ☐ It can attract natural enemies and regulate the microclimate, avoiding pests and diseases in the system
   ☐ It can increase the quality and nutrients in the soils
   ☐ It can make the system more resilient towards market changes
   ☐ It is more complicated to sell/ make profits out of two or more different products
   ☐ It can complicate my production system
   ☐ I don't have experience with it and I can be risking my income

*QUESTIONS JUST FOR FARMERS THAT WORK WITH AGROFOREST SYSTEMS*

6. **What are the main reasons for you to work with agroforest systems compared to conventional/monoculture farming system? Which benefits do you get from practicing this land-use system?**

   *Please mark with an X the main reasons:*

   ☐ Higher income generation
   ☐ The ability to sell the products at a higher unit price/price per kg (due to its organic origins, better quality of the products or different certification mechanisms)

    ☐    Increases the land's quality and productivity (water retention, improvement of the soils, use of different levels of production, microclimate regulation, etc.)

    ☐    A system that can give you different products at different times of the year

    ☐    More products and food for self sufficiency

    ☐    Agricultural system that is more resilient to climatic, biological (diseases) or market impairs

    ☐    It helps in conserving/increasing the biodiversity of flora and fauna in the region

    ☐    Other important reasons: _________________________________________________

**7. What are the disadvantages/most limiting factors when working with agroforest systems?**

*Please mark with an X the main disadvantages/limiting factors:*

    ☐    The long time one has to wait to make profits out of the woody perennials

    ☐    Reduction in yield of the main agricultural crop

    ☐    The logistical difficulty of managing and harvesting two (or more) plant species in the same area

    ☐    The difficulty to sell diverse products in the market

    ☐    Uncertainty towards the rights to own the land in the long term

    ☐    It attracts unwanted pests, insects, animals or other biological diseases

    ☐    Other important disadvantages/limitations: _________________________________

*QUESTIONS JUST FOR FARMERS THAT DON'T WORK WITH AGROFOREST SYSTEMS*

**8. If you could plant more woody perennials (such as trees, shrubs, palms, bamboos, etc.) in your field and thus potentially get benefits from this diversified agricultural system, but you would also be taking a risk in changing the dynamics of your land, would you consider implementing this kind of system?**

    ☐    Yes

    ☐    No

**8.1 If your answer was <u>yes</u>, which benefits would you be expecting to get from your farm diversified with woody perennials (such as trees, shrubs, palms, bamboos, etc.)?**

*Please mark with an X the main expected benefit<u>s</u>:*

    ☐    Higher income generation

    ☐    The ability to sell the products at a higher unit price/price per kg (due to its organic origins, better quality or different certification mechanisms)

    ☐    Increases the land's quality and productivity (water retention, improvement of the soils, use of different levels of production, microclimate regulation, etc.)

    ☐    A system that can give you different products at different times of the year

    ☐    More products and food for self sufficiency

    ☐    Agricultural system that is more resilient to climatic, biological (diseases) or market impairs

    ☐    It helps in conserving/increasing the biodiversity of flora and fauna in the region

    ☐    Other important reasons: _________________________________________________

**8.2 If your answer was <u>yes</u>, what do you think are the main barriers for diversifying your farm with woody perennials?**

*Please mark with an X the main barrier<u>s</u>:*

    ☐    Uncertainty towards how the system might work/it might fail

☐ Lack of successful models and knowledge in the region where I live to know how to make that transition

☐ The long time one has to wait to make profits out of the woody perennials

☐ Reduction in yield of the main agricultural crop

☐ The logistical difficulty of managing and harvesting two (or more) plant species in the same area

☐ The difficulty to sell diverse products in the market

☐ Uncertainty towards the rights to own the land in the long term

☐ It might attract unwanted pests, insects, animals or other biological diseases

☐ Other important disadvantages/limitations: _______________________________

**8.3 If your answer was <u>no</u>, what are the main reasons why you would not like to implement woody perennials in your farm?**

☐ Please mark with an X the main reason<u>s</u>:

☐ Uncertainty towards how the system might work/it might fail

☐ Lack of successful models and knowledge in the region where I live to know how to make that transition

☐ The long time one has to wait to make profits out of the woody perennials

☐ Reduction in yield of the main agricultural crop

☐ The logistical difficulty of managing and harvesting two (or more) plant species in the same area

☐ The difficulty to sell diverse products in the market

☐ Uncertainty towards the rights to own the land in the long term

☐ It might attract unwanted pests, insects, animals or other biological diseases

☐ Other important disadvantages/limitations: _______________________________

## Appendix B. Location of the Farms and Farm Sizes

**Table A1.** Location of conventional agriculture farms and farms sizes.

| Conventional Agriculture | | | | |
|---|---|---|---|---|
| **Farmer Number** | **State** | **Municipality** | **Size of the Farm (in ha)** | **Area Used for Agricultural Purposes (in ha)** |
| Farmer 1 | southern Bahia | Ipiaú | 7 | 1 |
| Farmer 2 | southern Bahia | Itacaré | 3 | 1 |
| Farmer 3 | southern Bahia | Itacaré | 3 | 2 |
| Farmer 4 | Espírito Santo | Barra de São Francisco | 1 | 1 |
| Farmer 5 | Espírito Santo | Barra de São Francisco | 7 | 1 |
| Farmer 6 | Espírito Santo | Barra de São Francisco | 8 | 3 |
| Farmer 7 | Espírito Santo | Conceição da Barra | 12 | 6 |
| Farmer 8 | Espírito Santo | Pedro Canário | 9 | |
| Farmer 9 | Espírito Santo | Pedro Canário | 10 | |
| Farmer 10 | Espírito Santo | Pedro Canário | 10 | |
| Farmer 11 | Espírito Santo | Pedro Canário | 10 | 3 |
| Farmer 12 | Espírito Santo | Pedro Canário | 10 | 9 |
| Farmer 13 | Espírito Santo | Pedro Canário | 24 | |
| Farmer 14 | Espírito Santo | São Gabriel da Palha | 5 | 3 |
| Farmer 15 | Espírito Santo | São Gabriel da Palha | 24 | 12 |
| Farmer 16 | Espírito Santo | São Gabriel da Palha | 28 | 21 |
| Farmer 17 | Espírito Santo | São Gabriel da Palha | 34 | 18 |
| Farmer 18 | Espírito Santo | Vila Pavão | 8 | 4 |

**Table A1.** *Cont.*

| Conventional Agriculture | | | | |
|---|---|---|---|---|
| **Farmer Number** | **State** | **Municipality** | **Size of the Farm (in ha)** | **Area Used for Agricultural Purposes (in ha)** |
| Farmer 19 | Espírito Santo | Vila Pavão | 26 | 15 |
| Farmer 20 | Espírito Santo | Vila Pavão | 34 | 27 |
| Farmer 21 | Espírito Santo | Vila Pavão | 45 | 22 |
| Farmer 22 | Minas Gerais | Rosário da Limeira | 1 | 1 |
| Farmer 23 | Minas Gerais | Rosário da Limeira | 3 | 3 |
| Farmer 24 | Minas Gerais | Rosário da Limeira | 4 | 4 |
| Farmer 25 | Minas Gerais | Rosário da Limeira | 5 | 4 |
| Farmer 26 | Minas Gerais | Rosário da Limeira | 9 | 7 |
| Farmer 27 | Minas Gerais | Rosário da Limeira | 12 | 3 |
| Farmer 28 | Minas Gerais | Rosário da Limeira | 15 | 15 |
| Farmer 29 | Minas Gerais | Antônio Prado de Minas | 26 | 24 |
| Farmer 30 | Minas Gerais | Espera Feliz | 2 | 1 |
| Farmer 31 | Minas Gerais | Eugenópolis | 19 | 9 |
| Farmer 32 | Minas Gerais | Eugenópolis | 22 | 18 |
| Farmer 33 | Minas Gerais | Eugenópolis | 24 | 19 |
| Farmer 34 | Minas Gerais | Eugenópolis | 29 | 3 |
| Farmer 35 | Minas Gerais | Eugenópolis | 48 | 33 |
| Farmer 36 | Minas Gerais | Laranjal | 24 | 20 |
| Farmer 37 | Minas Gerais | Laranjal | 25 | 22 |
| Farmer 38 | Minas Gerais | Laranjal | 35 | 30 |
| Farmer 39 | Minas Gerais | Leopoldina | 12 | 8 |
| Farmer 40 | Minas Gerais | Miradouro | 7 | 3 |
| Farmer 41 | Minas Gerais | Muriaé | 5 | 3 |
| Farmer 42 | Minas Gerais | Muriaé | 6 | 3 |
| Farmer 43 | Minas Gerais | Muriaé | 14 | 8 |
| Farmer 44 | Minas Gerais | Muriaé | 15 | 8 |
| Farmer 45 | Minas Gerais | Muriaé | 22 | 16 |
| Farmer 46 | Minas Gerais | Muriaé | 30 | 24 |
| Farmer 47 | Minas Gerais | Muriaé | 30 | 10 |
| Farmer 48 | Minas Gerais | Muriaé | 30 | 3 |
| Farmer 49 | Minas Gerais | Muriaé | 60 | 48 |
| Farmer 50 | Minas Gerais | Pedra Dourada | 2 | 2 |
| Farmer 51 | Minas Gerais | Pedra Dourada | 6 | 4 |
| Farmer 52 | Minas Gerais | Rosário da Limeira | 4 | 3 |
| Farmer 53 | Minas Gerais | Rosário da Limeira | 4 | 2 |
| Farmer 54 | Minas Gerais | Rosário da Limeira | 4 | 3 |
| Farmer 55 | Minas Gerais | Rosário da Limeira | 6 | 5 |
| Farmer 56 | Minas Gerais | Rosário da Limeira | 6 | 2 |
| Farmer 57 | Minas Gerais | Rosário da Limeira | 10 | 2 |
| Farmer 58 | Minas Gerais | Rosário da Limeira | 12 | 6 |
| Farmer 59 | Minas Gerais | Rosário da Limeira | 12 | 6 |
| Farmer 60 | Minas Gerais | Rosário da Limeira | 12 | 4 |
| Farmer 61 | Minas Gerais | Rosário da Limeira | 15 | 15 |
| Farmer 62 | Minas Gerais | Rosário da Limeira | 15 | 3 |
| Farmer 63 | Minas Gerais | Rosário da Limeira | 36 | 30 |
| Farmer 64 | Minas Gerais | Rosário da Limeira | 57 | 9 |

**Table A2.** Location of agroforestry farms and farms sizes.

| Agroforestry | | | | |
|---|---|---|---|---|
| **Farmer Number** | **State** | **Municipality** | **Size of the Farm (in ha)** | **Area Used for Agricultural Purposes (in ha)** |
| Farmer 1 | southern Bahia | Ilhéus | 12 | 7 |
| Farmer 2 | southern Bahia | Ilhéus | 12 | 8 |
| Farmer 3 | southern Bahia | Itacaré | 1 | 1 |
| Farmer 4 | southern Bahia | Itacaré | 2 | 1 |
| Farmer 5 | southern Bahia | Itacaré | 2 | 1 |
| Farmer 6 | southern Bahia | Itacaré | 3 | 2 |
| Farmer 7 | southern Bahia | Itacaré | 3 | 2 |
| Farmer 8 | southern Bahia | Itacaré | 3 | 2 |
| Farmer 9 | southern Bahia | Itacaré | 3 | 2 |
| Farmer 10 | southern Bahia | Itacaré | 3 | 2 |
| Farmer 11 | southern Bahia | Itacaré | 3 | 2 |
| Farmer 12 | southern Bahia | Itacaré | 4 | 3 |
| Farmer 13 | southern Bahia | Itacaré | 5 | 2 |
| Farmer 14 | southern Bahia | Itacaré | 5 | 3 |
| Farmer 15 | southern Bahia | Itacaré | 5 | 3 |
| Farmer 16 | southern Bahia | Itacaré | 5 | 4 |
| Farmer 17 | southern Bahia | Itacaré | 6 | 5 |
| Farmer 18 | southern Bahia | Itacaré | 7 | 3 |
| Farmer 19 | southern Bahia | Itacaré | 8 | 4 |
| Farmer 20 | southern Bahia | Itacaré | 8 | 6 |
| Farmer 21 | southern Bahia | Itacaré | 8 | 4 |
| Farmer 22 | southern Bahia | Itacaré | 9 | 4 |
| Farmer 23 | southern Bahia | Itacaré | 9 | 4 |
| Farmer 24 | southern Bahia | Itacaré | 10 | 8 |
| Farmer 25 | southern Bahia | Itacaré | 10 | 8 |
| Farmer 26 | southern Bahia | Itacaré | 10 | 8 |
| Farmer 27 | southern Bahia | Itacaré | 10 | 6 |
| Farmer 28 | southern Bahia | Itacaré | 10 | 3 |
| Farmer 29 | southern Bahia | Itacaré | 12 | 5 |
| Farmer 30 | southern Bahia | Itacaré | 13 | 3 |
| Farmer 31 | southern Bahia | Itacaré | 16 | 5 |
| Farmer 32 | southern Bahia | Itacaré | 17 | 10 |
| Farmer 33 | southern Bahia | Itacaré | 17 | 15 |
| Farmer 34 | southern Bahia | Itacaré | 18 | 9 |
| Farmer 35 | southern Bahia | Itacaré | 20 | 10 |
| Farmer 36 | southern Bahia | Itacaré | 22 | 3 |
| Farmer 37 | southern Bahia | Itacaré | 24 | 9 |
| Farmer 38 | southern Bahia | Itacaré | 26 | 13 |
| Farmer 39 | southern Bahia | Itacaré | 40 | 30 |
| Farmer 40 | southern Bahia | Maraú | 20 | 16 |
| Farmer 41 | southern Bahia | Uruçuca | 5 | 2 |
| Farmer 42 | southern Bahia | Uruçuca | 5 | 2 |
| Farmer 43 | southern Bahia | Uruçuca | 6 | 3 |
| Farmer 44 | Espírito Santo | Barra de São Francisco | 23 | 15 |
| Farmer 45 | Espírito Santo | Barra de São Francisco | 30 | 7 |
| Farmer 46 | Espírito Santo | Nova Venceia | 9 | 5 |
| Farmer 47 | Espírito Santo | Nova Venceia | 10 | 3 |
| Farmer 48 | Espírito Santo | Nova Venceia | 25 | 19 |
| Farmer 49 | Espírito Santo | Nova Venceia | 27 | 15 |
| Farmer 50 | Espírito Santo | Nova Venceia | 51 | 50 |
| Farmer 51 | Espírito Santo | Pedro Canário | 9 | 7 |
| Farmer 52 | Espírito Santo | São Gabriel da Palha | 4 | 2 |
| Farmer 53 | Minas Gerais | Miraí | 36 | 6 |
| Farmer 54 | Minas Gerais | Espera Feliz | 8 | 6 |

**Table A2.** *Cont.*

| Agroforestry | | | | |
|---|---|---|---|---|
| **Farmer Number** | **State** | **Municipality** | **Size of the Farm (in ha)** | **Area Used for Agricultural Purposes (in ha)** |
| Farmer 55 | Minas Gerais | Eugenópolis | 14 | 12 |
| Farmer 56 | Minas Gerais | Miradouro | 5 | 3 |
| Farmer 57 | Minas Gerais | Muriaé | 1 | 1 |
| Farmer 58 | Minas Gerais | Muriaé | 6 | 4 |
| Farmer 59 | Minas Gerais | Muriaé | 7 | 4 |
| Farmer 60 | Minas Gerais | Muriaé | 12 | 4 |
| Farmer 61 | Minas Gerais | Muriaé | 12 | 7 |
| Farmer 62 | Minas Gerais | Muriaé | 18 | 17 |
| Farmer 63 | Minas Gerais | Muriaé | 27 | 2 |
| Farmer 64 | Minas Gerais | Muriaé | 28 | 20 |
| Farmer 65 | Minas Gerais | Muriaé | 87 | 5 |
| Farmer 66 | Minas Gerais | Rosário da Limeira | 1 | 1 |
| Farmer 67 | Minas Gerais | Rosário da Limeira | 2 | 1 |
| Farmer 68 | Minas Gerais | Rosário da Limeira | 3 | 2 |
| Farmer 69 | Minas Gerais | Rosário da Limeira | 6 | 6 |
| Farmer 70 | Minas Gerais | Rosário da Limeira | 6 | 5 |
| Farmer 71 | Minas Gerais | Rosário da Limeira | 15 | 10 |
| Farmer 72 | Minas Gerais | Rosário da Limeira | 15 | 5 |
| Farmer 73 | Minas Gerais | Rosário da Limeira | 22 | 15 |
| Farmer 74 | Minas Gerais | São Francisco do Glória | 30 | 13 |
| Farmer 75 | Minas Gerais | São Francisco do Glória | 66 | 39 |

## Appendix C. Benefits of Working with Agroforestry According to the Three Analyzed States

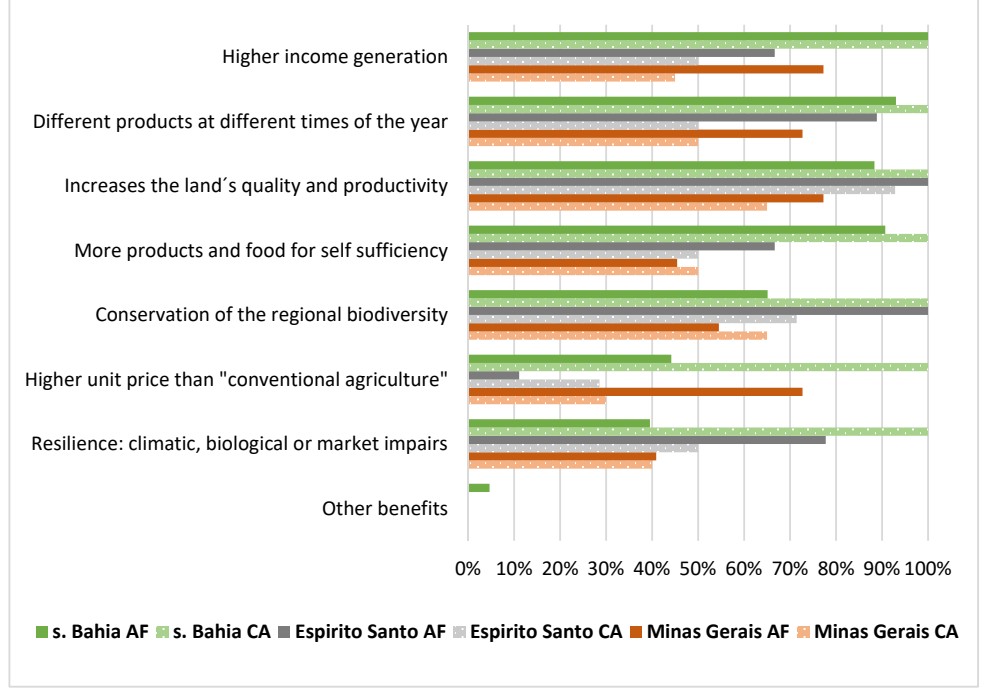

**Figure A1.** Main reasons and benefits of working with agroforestry compared to conventional agriculture by agroforestry farmers from southern Bahia (n = 43), Espírito Santo (n = 9), and Minas Gerais (n = 22). The expected benefits of switching to agroforestry practices, mentioned by the conventional agriculture farmers from southern Bahia (n = 1), Espírito Santo (n = 14), and Minas Gerais (n = 20) that would be willing to switch to agroforestry practices in their lands.

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
