# Peer review of "Agroforestry as a Biodiversity Conservation Tool in the Atlantic Forest? Motivations and Limitations for Small-Scale Farmers to Implement Agroforestry Systems in North-Eastern Brazil"

_sustainability, doi:10.3390/su11246932_

Round 1

Reviewer 2 Report

The paper "Agroforestry as a biodiversity conservation tool in the Atlantic Forest? Motivations and limitations for small-scale farmers to 

implement agroforestry systems in north-eastern Brazil" aims at invetsigating (i) the role of agroforestry in supporting biodiversity in the 

Atlantic forests and (ii) the motivations and limitations of farmers in implementing agroforestry in the area via surveying farmers.

Besides supporting the answer to the research questions of the paper, other useful insights emerge from the survey 

can be used in research in the field, especially when developing decision support tools (e.g., farmer like to have wild animal in their land), as well as 

defining and developping governamental initiatives to support environmental conservation (e.g., awareness vs participation to governamental support).

Despite the positive outcome regarding this study, I reported here below some specific comments that can hopefully improve the content and readability

of the manuscript. In particular, I suggest to the authors to re read the text and try to always be coherent with the two-fold aim of the paper when

presenting results and discussing them.

Specific comments:

- limitation of methods (survey) are clearly stated: the sample in not homogeneously distributed among geogr. areas

- Fig 3 - I would crop x axes of the bar chart to 30%

- better highlights the two-fold nature of the paper, from the aim, through results, to the discussion: (i) understand if agrof is supporting biodiversity

(ii) understand challenges and benefits of agroforestry perceived by farmers.

- I would move case study in material and methods; or avoiding to give a title to the paragraph and leave it in the introduction.

- Maybe few more information about the questionnaire can be useful (number of questions; number of questions per topic...).  

- Line 344-345: "they alone cannot provide a good enough habitat for these species and other local biodiversity." Is this an outcome of this study?

If yes, can you clarify which result support this sentence? If not, could you please add a reference?

- in the discussion paragraph "Farmers willingness to support biodiversity conservation through agroforestry" the authors can maybe discuss the results 

showed in Figure 5 (the awareness rate is much higher than participation one; so even if farmers know about existing support mechanisms, they do not participate)

- Line 390 and 395: isn't it Figure 4 instead of Figure 5?

- LIne 409: "higher revenues than conventional agriculture" I would add "at least in the medium-long term" (because in the short term, when perennial crops are growing

and non-productive, this cannot be verified)

- Line 440-443: I would underline also the importance of decision support tools and assessment frameworks

that can support farmers decisions via quantitative indicators. These will make farmers understanding

the economic, social and environmental performances of different alternative agricultural systems; and consequently also

understanding the opportunities brought by support initiatives like PES. An example is a recent study that I strongly suggest reference "Recanati, F., & Guariso, G. (2018). An optimization model for the planning of agroecosystems: Trading off socio-economic feasibility and biodiversity. Ecological Engineering117, 194-204. https://doi.org/10.1016/j.ecoleng.2018.03.010".

Round 2

Reviewer 1 Report

This version is much improved, and the thoughtful responses of the authors to my earlier comments appreciated. Though all of my minor recommendations were addressed, a few major concerns remain before I would recommend it for publication.

Major issue:

I remain convinced that the dual nature of this paper is unsatisfactory, specifically that the “literature review” presented as an equal research focus is not significant enough to be published as a research result. These 18 papers can be used in the introduction or background sections of this manuscript in order to motivate the research, but they do not themselves represent a research result. To meet this standard, a literature review should contain the criteria summarized here: https://research.library.gsu.edu/c.php?g=115595&p=754162. As presented in this paper, the 18 papers cited merely support the idea that the three focal species use cabrucas as habitat, though this use is predicated on the presence of nearby intact forest over the longer term. The inclusion of this literature as a research result in this manuscript is further weakened as the survey conducted by the authors does not collect data on canopy connectivity, hunting pressure, dog presence, or proximity to intact forest (the principle factors noted by the papers as important in determining conservation potential). As such, the methods, results, discussion, and conclusion sections must be revised to reflect only the novel data and results collected for this study. The 18 papers can be cited in the introduction or background as motivation for study site selection and the focus on agroforestry as a conservation tool.

After making these modifications, the survey data will take on greater prominence. To make this contribution sufficiently interesting for publication, the analysis should either (copied from my initial review comments):

Analyze the responses along axes other than agroforestry-conventional. For example along geographic boundaries, biodiversity boundaries, and cocoa-cattle-crop boundaries (those with cabruca or cocoa plantations versus those who have most of their land in pasture versus those who have mostly cassava/coconut/palm or some); and/or Use statistical analysis to assess if these differences between conventional and agroforestry farmers might be expected from chance in samples this small

Minor issues:

As written in my initial review, I would especially caution recommendations relating to PES and other programs, which are known locally by different names within Brazil. This is especially true for the PES-Atlantic Forest Law. I’ve never heard of this program, after working in southern Bahia on a PES. I also can’t find it referenced in the citation you provide. Nowhere does the Lei da Mata Atlantica appear near a pagamento por serviço ambiental. Does this refer to the Bolsa Verde or the Bolsa Floresta? In any case, unless the language of the survey used the proper name for the program, my guess is that people know about the program at higher rates and simply didn’t recognize the name used in the survey. I would either take out these data from your manuscript, or discuss this more thoroughly. As it stands, I think it likely presents a spurious result. Remove the “(Fig. 1)” references on lines 91, 92, 102, and 168. The first three of these claims are supported by the text well enough, and are not readily apparent in the figure. The last one should be shown in the figure, but is not. At the end of each section in the introduction, the gaps in the literature reviewed should be highlighted. This way, the papers aims will make better sense. The way it reads now, the introduction seems to already answer the questions posed as the aims of the presented work. Lines 132-136 describe the gap in farmer motivations/ barriers. This belongs earlier in the paper. In reference to papers that look at barriers and motivations to the use of agroforestry in Bahia, I recommend: McGinty, Meghan M., Mickie E. Swisher, and Janaki Alavalapati. "Agroforestry adoption and maintenance: self-efficacy, attitudes and socio-economic factors." Agroforestry systems2 (2008): 99-108. The descriptions of the Bahia SR are not consistent. Lines 85-88 should describe it as stretching along the coast of Brazil from Sergipe in the North through Bahia and ES in the south, extending inland up to 100 miles. Western BA is not included. Nor is any part of MG, though I think the authors mean to write eastern rather than western MG. There is a map on p. 49 of the cited reference that shows the region clearly. This will help to clarify that the study sites in MG described on lines 167-168 are indeed adjacent to the Bahia SR. It would be helpful to see a table of the conventional and agroforestry respondents from the three regions, including other axes, e.g. farm size. Based on information in lines 236-239 Combine lines 199-201 and 211-213 Lines 229-230 indicate surveys were conducted face to face, but earlier, it is said that surveys were “sent” to respondents. Please clarify. Move the lines 314-321 earlier.

Round 3

Reviewer 1 Report

I appreciate the authors’ responses to previous comments and the effort they put forward to address them. As a result, I find the current version much clearer and more impactful.

I made a few comments throughout the paper (attached) for minor revisions.

The only content comments I have relate to the discussion and conclusions sections:

The discussion is the place to highlight this paper’s contributions and implications in the literature and for practice. Entire paragraphs that refer only to work done by others does not belong in the discussion. This work is only relevant if it gives context and meaning to your results. This applies to the 2nd paragraph of section 4.2.1, the 2nd and 3rd paragraphs of 4.2.2, the 2nd paragraph of 4.3.1, and the 3rd, 4th, and 5th paragraphs of section 4.3.2. A citation is needed for the claim made that ends on line 423. Lines 443-448 need to be clarified. New major citations should not be introduced in the discussion- e.g., citations 36, 37 and 38, and 40. If they are important enough to the context of your work, they should be introduced in the introduction. Remove language from the conclusion that doesn’t come from your work (e.g., lines 515-518).
